# A Randomized Comparison of Multimodal Analgesia and Fentanyl-Based Patient-Controlled Analgesia in Women Undergoing Robot-Assisted Bilateral Axillary Breast Approach Thyroidectomy

**DOI:** 10.3390/jcm13030702

**Published:** 2024-01-25

**Authors:** Na-Young Tae, Jin Wook Yi, Jong-Kwon Jung, Junhyung Lee, Sooman Jo, Hyunzu Kim

**Affiliations:** 1Department of Anesthesiology and Pain Medicine, Inha University College of Medicine, Incheon 22212, Republic of Korea; taena0@gmail.com (N.-Y.T.); ckchung@inha.ac.kr (J.-K.J.); ljh112700@gmail.com (J.L.); ssoosoo1@naver.com (S.J.); 2Department of Surgery, Inha University College of Medicine, Incheon 22212, Republic of Korea; jinwook.yi@inha.ac.kr

**Keywords:** thyroid cancer, multimodal analgesia, fentanyl, robot-assisted bilateral axillary breast approach

## Abstract

Opioid-free multimodal analgesia (MMA) emerges as a preferable approach for postoperative pain management compared to opioid-based patient-controlled analgesia (PCA) in robot-assisted bilateral axillary breast approach thyroidectomy, a procedure commonly undergone by young female patients. We compared the analgesic efficacy and other recovery profiles between MMA and PCA. In total, 88 female patients were administered fentanyl-based PCA or the combination of lidocaine continuous infusion and nefopam injection before recovery from general anesthesia. The visual analog scale score of postoperative pain was assessed at the post-anesthesia care unit and at 6, 12, and 24 h after the termination of surgery. Postoperative nausea and vomiting (PONV), rescue analgesic and anti-emetic agents, recovery profiles, and adverse events were also compared. The median numeric rating scores on postoperative pain at 6 h after recovery from general anesthesia were three in both groups, with no significant difference between the groups at any time point. The PONV incidence was significantly higher in the PCA group than in the MMA group. The combination of systemic lidocaine infusion and nefopam injection has an analgesic effect equivalent to that of fentanyl-based PCA without PONV.

## 1. Introduction

Thyroid cancer is one of the most prevalent cancers affecting young women [1]. The cosmetic outcomes of surgical scars play a pivotal role in determining post-thyroidectomy quality of life, enabling most patients to swiftly resume their daily activities. Although traditional open thyroidectomy has proven to be effective and safe, alternative minimally invasive thyroid surgeries such as endoscopic and robot-assisted thyroidectomy have increased in popularity [2]. In particular, the robot-assisted bilateral axillary breast approach (BABA) is currently the most widely used method, with approximately 2300 robot-assisted BABA thyroidectomies performed in South Korea in 2022 alone.

Multimodal analgesia (MMA) consists of a combination of non-opioid analgesics with different targets for additive or synergistic effects to provide effective analgesia against postoperative pain. Physiological and pharmacologic benefits are maximized, and adverse effects are minimized. Conventional use of opioids during the perioperative period causes several complications. Postoperative nausea and vomiting (PONV) is not only the most common adverse event in the postoperative period [3] but is also a highly distressing experience, especially for young women, and severely reduces patient satisfaction [4]. In terms of the immunologic system, almost all opioids suppress cell-mediated immunity, which is essential for detecting and eliminating the remaining malignant cells after resecting primary cancer [5,6]. As a result, the risk of recurrence or distant metastasis can increase with poor patient prognosis [7]. In addition, excessive use of opioids involves the risk of abuse or addiction [8]. Therefore, the opioid-sparing effects of MMA have been observed, reducing opioid consumption and its adverse effects.

Postoperative pain following robot-assisted BABA has been reported to be comparable to that of traditional open thyroidectomies [9,10], with opioid-based analgesia conventionally used in many cases. Therefore, it is imperative to assess the analgesic efficacy and other characteristics of opioid-free MMA after robot-assisted BABA, particularly in female patients who are more vulnerable to the complications of opioid-based analgesia. We hypothesized that our institution’s MMA regimen would result in analgesic efficacy equivalent to that of opioid-based PCA in robot-assisted BABA thyroidectomy. Thus, in this prospective study, we compared the analgesic efficacy and other recovery profiles of the opioid-based PCA and opioid-free MMA groups.

## 2. Materials and Methods

This study was conducted according to the guidelines of the Declaration of Helsinki and was registered with the Clinical Research Information Service (identifier: KCT0006638) after the protocol was approved by the Institutional Review Board of Inha University Hospital (Incheon, Republic of Korea, protocol code INHA-2021-02-005; date of approval 26 April 2021). Written informed consent was obtained from participants who were female patients aged ≥ 20 years, with the American Society of Anesthesiologists Classification of 1–3, scheduled for BABA robotic thyroidectomy for thyroid cancer. Patients were equally randomized to one of two groups: the PCA group with fentanyl or the opioid-free multimodal analgesia group (MMA Group) according to the list prepared through the Internet Randomization Site (https://www.sealedenvelope.com/simple-randomiser/v1/lists, accessed on 5 January 2021). Patients were excluded from the study if they had chronic pain syndrome, psychologic or neurologic dis-eases, arrhythmia, hepatic or renal diseases, or diseases of the respiratory system. Trial Registration Number: The Korean Clinical Research Information Service (CRIS) Registration Number: KCT0006638.

No patient was premedicated upon arrival at the operating room. Patients were monitored via non-invasive blood pressure assessment, pulse oximetry, electrocardiography, electroencephalogram (EEG)-based anesthetic depth monitoring, SedLine (Masimo Corporation, Irvine, CA, USA), and neuromuscular monitoring via Twitch View (Blink Device Company, Seattle, WA, USA). General anesthesia was induced with propofol at an effect-site concentration (Ce) of 5 µg/mL according to Schnider’s model and remifentanil was administered at a Ce of 3 ng/mL according to Minto’s pharmacokinetic model. The propofol and remifentanil infusion rates were controlled using a target-controlled infusion (TCI) of an Agilia^®^ SP TIVA infusion pump (Fresenius Vial S.A.S., Le Grand Chemin, Brezins, France). After initiating neuromuscular monitoring, 1 mg/kg rocuronium was administered. In the MMA group, 1 mg/kg lidocaine was administered and then continuous 3 µg/kg/min lidocaine infusion was initiated until surgical suture completion. After neuromuscular blockade was confirmed by 0 TOF, endotracheal intubation was performed using an internal diameter 6.0 mm EMG tube (NIM Trivantage EMG tube; Medtronic, Minneapolis, MN, USA). The proper positioning of electrodes on the endotracheal tube into the vocal cord was confirmed using the NIM Nerve Monitoring System (Medtronic, Minneapolis, MN, USA). Mechanical ventilation was adjusted to maintain end-tidal CO_2_ pressure at 30–35 mmHg at FiO_2_, 0.4. The Ce of propofol TCI was maintained using the Patient State Index (PSI) value of Sedline to ensure it remained between 30 and 40. The Ce of remifentanil TCI was controlled to achieve blood pressure and heart rate within 20% of preoperative baseline values. Rocuronium was infused continuously to achieve 1 Train of Four (TOF). After inducing anesthesia, 0.075 mg palonosetron was administered to both groups. When the surgeon started to suture the strap muscles, 20 mg of nefopam was mixed with 100 mL of normal saline and administered for 15 min in the MMA group. In the PCA group, 1 mcg/kg of fentanyl was injected, the PCA device was initiated, and the drugs were infused continuously. IV-PCA consisted of fentanyl (20 μg/kg) at a basal infusion rate of 0.5 mL/h (fentanyl 0.17 μg/kg) and a bolus of 0.5 mL (fentanyl 0.17 μg/kg) with a lockout interval of 15 min. Propofol TCI was stopped when the robotic system was de-docked, and remifentanil TCI was stopped at the end of the surgery in both groups. Simultaneously, another anesthesiologist, blinded to the patient’s group, performed the remaining recovery processes. Anesthesiologists were assigned to the recovery period with no information about this study at all and only had the protocol about recovery from general anesthesia explained to them by the researcher. After recovery of neuromuscular function was confirmed by 4 TOF, 2 mg/kg sugammadex was administered. Mechanical ventilation was then converted to manual ventilation. The patient was not disturbed except for continuous verbal requests to open her eyes. When the patients opened their eyes, deep breathing was encouraged, and after confirmation of the ability to maintain spontaneous respiration, the patient’s trachea was extubated after cuff deflation. After transferring the patient to the post-anesthesia care unit (PACU), 30 mg ketorolac was administered to the MMA group, and 0.5 mg/kg of fentanyl was administered to the PCA group if postoperative pain was above 5 points on the visual analog scale (VAS). If the patient had postoperative nausea and vomiting (PONV), dexamethasone (5 mg) was administered. In the general ward, patients in both groups received 1000 mg of acetaminophen and 1 mg of granisetron IV bolus twice daily on the day of surgery.

Pain scores were assessed and recorded in the PACU and at 6, 12, and 24 h after anesthesia using the VAS. Systolic and diastolic blood pressure and heart rate were recorded before and after intubation and extubation. The severity of cough was graded on a four-level scale adapted from the scale of Minogue and colleagues: 0, no cough; 1, single cough; 2, more than one episode of non-sustained cough; 3, sustained and repetitive cough. In the PACU, the sedation scale at 10 min after admission, nausea, vomiting, consumption of rescue medication for postoperative pain or PONV, Aldrete score, and stay duration were recorded. Sedation was graded according to the following scoring guidelines: 0, unresponsive to any stimulus; 1, responsive to a loud voice or tactile stimulus; 2, responsive to a voice at an ordinary pitch; 3, alert and responsive. In the general ward, PONV was recorded at 6, 12, and 24 h after anesthesia. Patient satisfaction was evaluated using the 40-item quality-of-recovery score (QoR-40) on the day after surgery.

The primary outcome was the VAS score of postoperative pain measured 6 h after anesthesia. Based on a previous study comparing pain scores in robotic thyroidectomy, non-inferiority trials were set with a standard deviation of 15 points and a non-inferiority margin of 10 points. The value for the non-inferiority test was set to 0.05, and the power (1-β) was set to 0.9. The minimum number of samples required was 39 per group. Considering a dropout rate of 10%, a total sample size of 88 was needed. All results are expressed as mean ± standard deviation or median (quartile) and percentage using SPSS 26.0 (SPSS Inc., Chicago, IL, USA). The 90% confidence interval of the difference in the primary outcome between both groups was used to determine whether the value was included in the non-inferiority limit. After testing for normally distributed data using the Kolmogorov–Smirnov and Shapiro–Wilk tests, continuous variables were analyzed using an independent *t*-test or Mann–Whitney U test. Categorical variables were analyzed using a chi-squared test or Fisher’s exact test. A *p*-value of < 0.05 was considered statistically significant. Per-protocol analysis was performed to manage missing values.

## 3. Results

Of the 88 patients enrolled, 1 patient was excluded from the final analysis. Figure 1 illustrates the patient enrolment process and study flow. Patient characteristics are presented in Table 1. No significant differences in demographic and surgical characteristics were observed between the two groups.

Figure 2 shows that no significant differences existed between the groups in pain scores at any time. In both groups, the median VAS score was five in the PACU and three after 24 h in the general ward. One patient in the MMA group required additional analgesics (ketorolac), and one patient in the PCA group required additional fentanyl administration. The hemodynamic changes before and after intubation and extubation are summarized in Table 2. 

The systolic pressure immediately after intubation was lower in the MMA group than in the PCA group. Table 3 presents the intraoperative profiles, including the requirement of anesthetic agents and durations. 

No significant differences existed in the profiles of the recovery stage and PACU, including awareness time and cough severity, between the groups (Table 4). 

The profiles of postoperative complications in the wards are summarized in Table 5. 

PONV incidence was significantly higher in the PCA group than in the MMA group. The quality-of-life scores on the day after the operation were comparable in both groups. In addition, two patients in the MMA group requested additional analgesics in the ward 24 h postoperatively. In addition, two patients in the PCA group requested the discontinuation of PCA infusion due to side effects. These situations were considered assigned treatment failures and were comparable between groups. The incidence of headaches and other complications was comparable between the groups.

## 4. Discussion

This prospective study compared the analgesic efficacy and other recovery profiles between the fentanyl-based PCA and opioid-free MMA groups. Although various regimens have been suggested, we investigated an MMA regimen consisting of continuous lidocaine infusion and nefopam. Patients who received opioid-free MMA had an analgesic efficacy equivalent to that of fentanyl-based PCA, whereas the PONV incidence was significantly lower in the MMA group. Thus, continuous lidocaine infusion and nefopam can replace opioid-based PCA.

Robot-assisted BABA thyroidectomy is increasing yearly along with transoral approaches. This trend is expected to continue for a while as the postoperative scar is barely noticeable and postoperative recovery is improving [11]. Robot-assisted BABA thyroidectomy requires wide subcutaneous detachments for the space where endoscopic instruments can move without colliding [12]. Widespread and intense pain occurs at the flap site on the upper chest [13]. In addition to this characteristic of surgical pain, another feature to consider for proper postoperative analgesia is demographic prevalence. Thyroid cancer is one of the most common cancers in young women who are critically concerned about PONV [14]. PONV interferes with appropriate postoperative analgesia and adversely affects the healing process of a surgical wound. Finally, this common adverse event reduces the quality of recovery and increases the length of stay in the PACU, hospitalization period, and medical costs [15]. Therefore, postoperative pain control regimens must take this into account, as most cases occur in young women who also experience surgical pain.

In this study, the MMA regimen controlled postoperative pain in the PACU and at 6, 12, and 24 h after thyroidectomy as effectively as fentanyl-based PCA. Postoperative pain after thyroidectomy is generally classified as mild-to-moderate surgical pain and is most intense within 6 h postoperatively [16]. There is a risk of providing opioid analgesics in excessive amounts or for a prolonged duration as PCA regimens cannot be sufficiently individualized. In particular, if the postoperative pain is mild-to-moderate or the patient has a risk of adverse effects from opioids, MMA without opioids may be a more appropriate strategy. In the general ward, two patients in the MMA group requested additional analgesics. We established this as treatment failure; however, it is impossible to conclude that the MMA approach has a weaker analgesic impact than the PCA method when less than 5% of all patients seek additional analgesic agents. Moreover, there was no statistical difference in postoperative pain scores between the two groups. Furthermore, patients in the PCA group received an additional fentanyl bolus via the PCA system’s pushing button. Both protocols in this study provided effective analgesia during the immediate postoperative period.

In this study, PONV incidence requiring anti-emetic agents in the ward was significantly lower in the MMA group than in the PCA group. In addition, two patients requested PCA discontinuation owing to PONV side effects, which can have a negative effect regarding cost and reduced satisfaction with perioperative care. Thus, our results indicate that multimodal pain management can minimize the adverse effects of opioid analgesics such as PONV.

In previous studies, continuous intraoperative lidocaine infusion has shown several advantages, such as postoperative analgesia and improved quality of recovery after robotic thyroidectomy [17]. Intravenous lidocaine also indirectly affects intestinal motility through pain reduction, opioid-sparing effect, and sympathetic blockade [18]. These beneficial effects explain the lower incidence of PONV in the MMA group. In addition, according to recent experimental studies, intravenous lidocaine can improve the outcomes of patients with cancer. The suggested mechanisms are that intravenous lidocaine directly blocks voltage-gated sodium channels and inhibits the secretion of inflammatory mediators, which is essential for the migration and distant metastasis of primary cancer cells [19,20]. Nefopam, which is a centrally acting analgesic with both supraspinal and spinal sites of action, inhibits serotonin, norepinephrine, and dopamine reuptake [21]. Nefopam acts as an *N*-methyl-d-aspartate receptor antagonist and affects the glutamatergic pathway via modulation of sodium channels. Through these mechanisms, nefopam can suppress the development of chronic pain or sensory disturbance as well as acute postoperative pain [22]. In the present study, the combination of intravenous lidocaine and nefopam was expected to reduce postoperative pain after robotic or endoscopy-assisted thyroidectomy, and effective multimodal analgesia without adverse effects was achieved.

Preventing cough during extubation of the endotracheal tube is also a concern in various surgical fields. Considerably severe cough during the recovery period after thyroidectomy increases the risk of postoperative bleeding [23]. To prevent extubation-related cough, medications, such as opioids [24], lidocaine [25], and dexmedetomidine [26], with diverse mechanisms, have been suggested. In this study, none of the patients in either group coughed severely at the moment of extubation. Thus, the effect of fentanyl in the PCA group was equivalent to that of lidocaine in the MMA group. Furthermore, no difference was observed in the incidence of recovery profiles between the two groups.

No significant difference was observed in postoperative complication incidence between the groups. No complications related to intravenous lidocaine infusion into the central nervous system, such as numbness, sensory disturbance, or dizziness, were reported. In addition, the low-dose lidocaine infusion was confirmed to be relatively safe, consistent with previous results. Nefopam has fewer side effects than other analgesics, including sweating, confusion, and tachyarrhythmia, which have rarely been reported [27]. These side effects were not observed in this study. Furthermore, eight patients in both groups complained of headaches, and no significant difference was observed between the groups. Epinephrine injection for building flaps up at the beginning of surgery is speculated to be the source of the headache.

This study has several limitations. First, owing to the technical features of our institute’s PCA system, data related to PCA consumption over time were not corrected. Additionally, according to the PONV prevention guidelines revised in 2019, three or more prophylactic medications are recommended if there are more than three risk factors. However, in this study protocol, two prevention methods were used, and no patient had PONV in the recovery room. Nonetheless, outbreaks occurred in the PCA group in the ward. Further research is warranted to comprehensively compare the effects of different MMA regimens or male patients for robotic-assisted thyroidectomy.

## 5. Conclusions

Opioid-based analgesia presents several potential drawbacks for young women undergoing robot-assisted BABA thyroidectomy; therefore, a potential enhancement in outcomes could be achieved by substituting opioid-based analgesia with opioid-free MMA. The combination of continuous lidocaine infusion and nefopam injection has been shown to effectively reduce acute postoperative pain compared with fentanyl-based PCA while significantly reducing the incidence of PONV in women undergoing robot-assisted BABA thyroidectomy. This MMA regimen can replace opioid-based PCA. More research is required to compare the effects of various MMA regimens for robotic-assisted thyroidectomy.

## Figures and Tables

**Figure 1 jcm-13-00702-f001:**
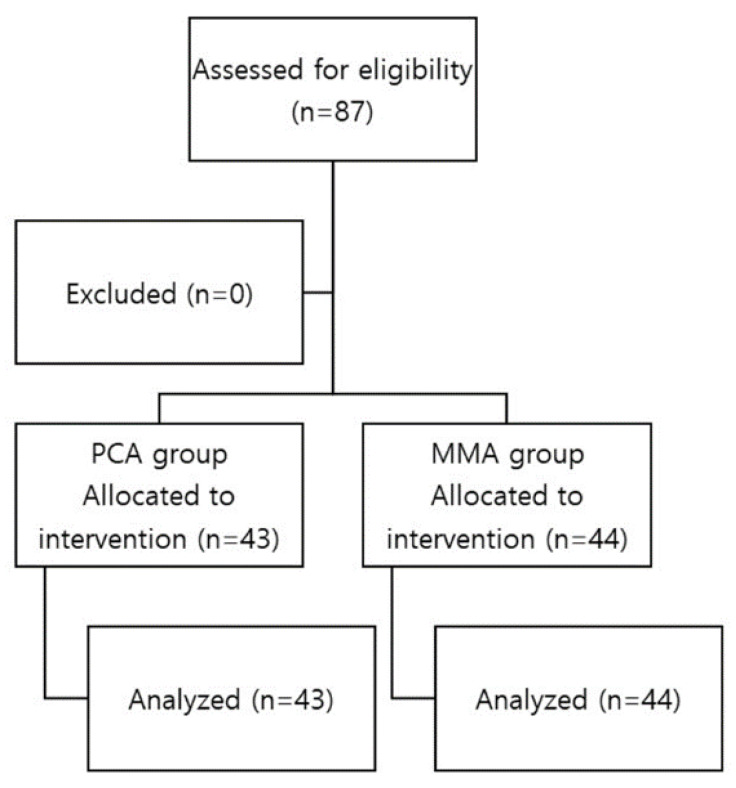
Enrolment and allocation of patients in the study.

**Figure 2 jcm-13-00702-f002:**
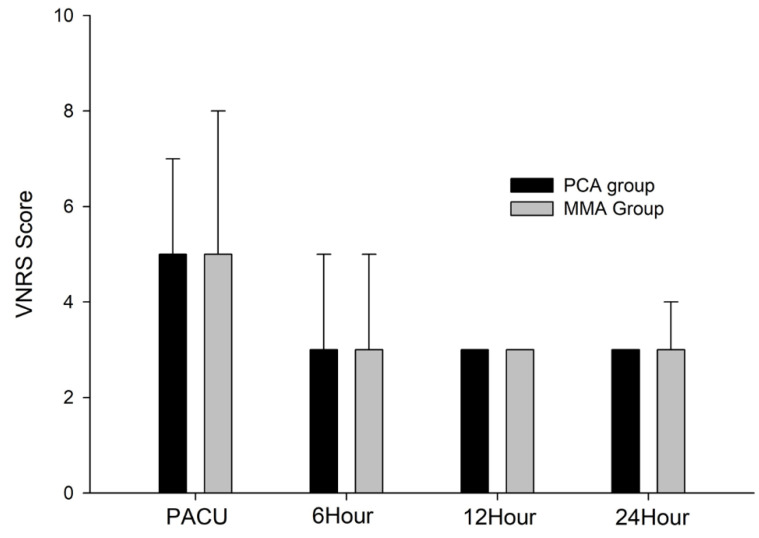
The median visual analogue scores in the post-anesthesia care unit and general ward in both groups. PCA, patient-controlled analgesia; MMA, multimodal analgesia; VNRS, visual numeric rating scale; *PACU*, post-anesthesia care unit.

**Table 1 jcm-13-00702-t001:** Distribution of demographic data between PCA and MMA groups.

	PCA Group (n = 43)	MMA Group (n = 44)	*p*-Value
Age (year)	48.1 ± 11.2	43.7 ± 11.8	0.78
Height (cm)	158.1 ± 6.4	158.6 ± 5.7	0.65
Weight (kg)	60.3 ± 10.1	64.2 ± 11.4	0.10
ASA class (I/II)	20/23	22/22	0.91

Values are given as mean ± standard deviation or number of patients; *ASA* class, American Society of Anaesthesiologists physical class.

**Table 2 jcm-13-00702-t002:** Hemodynamic changes before and after intubation and extubation between PCA and MMA groups.

		PCA Group (n = 43)	MMA Group (n = 44)	*p*-Value
Before intubation	SBP (mmHg)	141.6 ± 30.5	136.0 ± 28.7	0.40
	DBP (mmHg)	70.4 ± 13.3	67.9 ± 13.1	0.39
	Heart Rate (bpm)	69.6 ± 13.7	70.1 ± 12.6	0.84
After intubation	SBP (mmHg)	136.9 ± 21.1	122.0 ± 20.6	0.001 *
	DBP (mmHg)	72.6 ± 11.7	65.4 ± 11.5	0.05
	Heart Rate (bpm)	77.6 ± 13.7	72.6 ± 12.7	0.085
Before extubation	SBP (mmHg)	127.7 ± 24.8	136.9 ± 24.3	0.085
	DBP (mmHg)	64.6 ± 14.0	66.3 ± 12.5	0.53
	Heart rate (bpm)	75.3 ± 10.6	74.9 ± 15.3	0.88
After extubation	SBP (mmHg)	153.1 ± 24.3	161.5 ± 21.5	0.09
	DBP (mmHg)	77.3 ± 12.5	79.2 ± 10.5	0.44
	Heart Rate (bpm)	82.7 ± 12.7	84.3 ± 15.9	0.61

Values are given as mean ± standard deviation; *SBP*, systolic blood pressure; *DBP*, diastolic blood pressure.; * *p*-value between PCA group and MMA group, as compared by the paired *t*-test.

**Table 3 jcm-13-00702-t003:** Intraoperative profiles between PCA and MMA groups.

	PCA Group (n = 43)	MMA Group (n = 44)	*p* Value
Propofol requirement (mg)	8.7 ± 1.8 × 102	8.6 ± 1.5 × 102	0.77
Remifentanil requirement (μg)	9.2 ± 2.6 × 102	10.1 ± 2.7 × 102	0.11
MRND	3	4	0.98
Surgical duration (min)	128.8 ± 44.0	129.7 ± 51.2	0.94
Anesthetic duration (min)	173.8 ± 43.4	169.2 ± 53.2	0.66

Values are given as mean ± standard deviation or number of patients; *MRND*, Modified radical neck dissection

**Table 4 jcm-13-00702-t004:** Recovery profiles between PCA and MMA groups.

	PCA Group (n = 43)	MMA Group (n = 44)	*p* Value
Awareness time (s)	348.4 ± 177.7	392.7 ± 182.9	0.32
Coughing	0 (1)	0 (1)	0.12
Rescue analgesics	1 (2)	1 (2)	0.99
Rescue antiemetics	0 (0)	0 (0)	0.57
Sedation level	3 (0)	3 (0)	0.87
Aldete score	10 (1)	10 (1)	0.21
PACU duration (min)	52.3 ± 19.8	52.3 ± 21.6	0.99

Values are given as mean ± standard deviation or median (IQR); *PACU*, post-anesthesia care unit

**Table 5 jcm-13-00702-t005:** Postoperative profiles in the wards between PCA and MMA groups.

	PCA Group (n = 43)	MMA Group (n = 44)	*p* Value
PONV	15 (35%)	1 (0.25%)	>0.001
QoR-40	149.5 ± 25.8	156.2 ± 21.9	0.19
Headache	4 (9.3%)	4 (9%)	0.74
Failure	2 (4.6%)	2 (4.5%)	0.63

Values are given as mean ± standard deviation or number of patients (%); PONV, postoperative nausea and vomiting; QoR-40, 40-item quality of recovery score

## Data Availability

The data are not publicly available due to we didn’t get consent about disclose of data from the participants.

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
