# Peer review of "A Randomized Comparison of Multimodal Analgesia and Fentanyl-Based Patient-Controlled Analgesia in Women Undergoing Robot-Assisted Bilateral Axillary Breast Approach Thyroidectomy"

_jcm, 2024, doi:10.3390/jcm13030702_

Round 1
Reviewer 1 Report
Comments and Suggestions for Authors
Major Revisions:
- In Table 3, MRND is performed in 3 and 4 patients in the PCA and MMA groups, respectively. Please specify the number of other surgical procedures (unilateral lobectomy, total thyroidectomy) in both the PCA and MMA groups, as these procedures have different durations. Combining them for mean surgical duration might not be suitable.
- In lines 111-113, you mention that another anesthesiologist, blinded to the patient’s group, performed the remaining recovery processes. However, it seems challenging for this process to be blinded, given the visible PCA device in the PCA group and the absence of a PCA device (only one shot of nefopam) in the MMA group.
- Was it more challenging or easier for the MMA group to control the anesthetic depth during the operation compared to the PCA group? Did patients in the MMA group experience more bucking during surgery?
- Some surgeons apply ropivacaine injection to the skin flap to reduce postoperative pain in BABA thyroidectomy. Did you incorporate this technique into your MMA group?
- Could you discuss the medical cost on the patient’s side between the PCA and MMA groups? Which group has to pay more?
Minor Revisions:
- In line 113, prior to employing the abbreviation for the first time, kindly provide the full name "Train of Four (TOF)."
- In lines 177-178, please check the meaning of this sentence: “In addition, two patients in the PCA group requested PCA infusion due to side effects.” It seems that it should be "declined PCA infusion" rather than "requested PCA infusion."
Author Response
We have read the reports you have sent us regarding our manuscript “A randomized comparison of multimodal analgesia and fentanyl-based PCA in women undergoing robot-assisted bilateral axillary breast approach thyroidectomy” (Manuscript ID: jcm-2796545)
Thank you for your comments and guidance.
We have corrected the errors in our paper that the reports have pointed out, and have further refined the manuscript in accordance with the comments before resubmission. We have highlighted the revised portions with red font and responded to each comment in the reports below, which we have sent to the notes to the editor section.
Major Revisions:
- In Table 3, MRND is performed in 3 and 4 patients in the PCA and MMA groups, respectively. Please specify the number of other surgical procedures (unilateral lobectomy, total thyroidectomy) in both the PCA and MMA groups, as these procedures have different durations. Combining them for mean surgical duration might not be suitable.
Response: I fully understand your question. Actually, patients who underwent total thyroidectomy were determined to operate according to the results of frozen section biopsy during surgery. The surgeon already performed one thyroidectomy and waited for the result with the robot system docked in the surgical field. And after that according to the result of the frozen section biopsy and decided the other thyroidectomy. Accordingly, the difference in the operation time of unilateral or bilateral thyroidectomy was within 30 minutes, so we did not separate the results.
- In lines 111-113, you mention that another anesthesiologist, blinded to the patient’s group, performed the remaining recovery processes. However, it seems challenging for this process to be blinded, given the visible PCA device in the PCA group and the absence of a PCA device (only one shot of nefopam) in the MMA group.
Response: Thank you for your comment. There is not only one anesthesiologist who wakes up the patient, and they were put into the recovery period are only familiar with how to wake up patients without any information on this study. We apologize for the uncertain explanation on this part. We have added this content to the method section.
- Was it more challenging or easier for the MMA group to control the anesthetic depth during the operation compared to the PCA group? Did patients in the MMA group experience more bucking during surgery?
Response: Thank you for your comment. In the two groups, there was no significant difference in the consumption of anesthetic agents such as propofol and remifentanil used during surgery or various recovery profiles.
- Some surgeons apply ropivacaine injection to the skin flap to reduce postoperative pain in BABA thyroidectomy. Did you incorporate this technique into your MMA group?
Response: The surgeon of our institute didn’t use any local anesthetic agent in the surgical field. So no patients were administered ropivacaine injection.
- Could you discuss the medical cost on the patient’s side between the PCA and MMA groups? Which group has to pay more?
Response: Thank you for your comment about medical costs. In South Korea, public medical insurance guarantees most of the medications, especially for cancer patients, the actual patient's burden is less than 10%, so there is no significant difference between the two groups. Therefore, at the informed consent stage, the patients were explained and agreed to all of them to be included in the study.
Minor Revisions:
- In line 113, prior to employing the abbreviation for the first time, kindly provide the full name "Train of Four (TOF)."
Response: Thank you. We had added the full name missed out before the abbreviation.
- In lines 177-178, please check the meaning of this sentence: “In addition, two patients in the PCA group requested PCA infusion due to side effects.” It seems that it should be "declined PCA infusion" rather than "requested PCA infusion."
Response: Thank you for your valid comment. We had meant that two patients declined

Reviewer 2 Report
Comments and Suggestions for Authors
Dear authors,
The manuscript entitled A randomized comparison of multimodal analgesia and fentanyl-based PCA in women undergoing robot-assisted bilateral axillary breast approach thyroidectomy is an original article.
The authors compared the analgesic efficacy and other recovery profiles between opioid-based patient-controlled analgesia (PCA) and opioid-free multimodal analgesia (MMA) in female patients undergoing robot-assisted bilateral axillary breast approach (BABA) thyroidectomy for thyroid cancer. The study was conducted on 43 and 44 female patients with PCA and MMA respectively. The version of MMA was a combination of continuous intraoperative lidocaine infusion and nefopam
The article is well-written and in clear scientific language.
The methodology is clearly explained, with inclusion and exclusion criteria. The statistical analysis was rigorous.
The Tables and Figures are with a complete legend. However, in Figure 2 it would be useful to explain the legend for the “VNRS” score from the graph.
The results of the study concluded that the combination of continuous lidocaine infusion and nefopam injection reduced acute postoperative pain compared with fentanyl-based PCA, while significantly reducing the incidence of postoperative nausea and vomiting (PONV) in women undergoing robot-assisted BABA thyroidectomy.
The limitations are presented fairly. The study used two prevention methods instead of a minimum of three as usual guides for postoperative nausea and vomiting (PONV) prevention, and no patient had PONV in the PO room. I would mention that the patients were females exclusively.
References are relevant, but it could be useful to discuss potential allergic reactions to anesthetic drugs and the evaluation of this risk. The authors can use as a reference:
Dumitru M, Berghi ON, Taciuc IA, Vrinceanu D, Manole F, Costache A. Could Artificial Intelligence Prevent Intraoperative Anaphylaxis? Reference Review and Proof of Concept. Medicina (Kaunas). 2022 Oct 26;58(11):1530. doi: 10.3390/medicina58111530. PMID: 36363487; PMCID: PMC9694532.
The article is a good work with clinical applicability. It needs more clinical trials to confirm the results (on male and female patients) and, of course, other variants of MMA, also.
Author Response
We have read the reports you have sent us regarding our manuscript “A randomized comparison of multimodal analgesia and fentanyl-based PCA in women undergoing robot-assisted bilateral axillary breast approach thyroidectomy” (Manuscript ID: jcm-2796545)
Thank you for your comments and guidance.
//Dear authors,
The manuscript entitled A randomized comparison of multimodal analgesia and fentanyl-based PCA in women undergoing robot-assisted bilateral axillary breast approach thyroidectomy is an original article.
The authors compared the analgesic efficacy and other recovery profiles between opioid-based patient-controlled analgesia (PCA) and opioid-free multimodal analgesia (MMA) in female patients undergoing robot-assisted bilateral axillary breast approach (BABA) thyroidectomy for thyroid cancer. The study was conducted on 43 and 44 female patients with PCA and MMA respectively. The version of MMA was a combination of continuous intraoperative lidocaine infusion and nefopam
The article is well-written and in clear scientific language.
The methodology is clearly explained, with inclusion and exclusion criteria. The statistical analysis was rigorous.
The Tables and Figures are with a complete legend. However, in Figure 2 it would be useful to explain the legend for from the graph.
Response: Thank you for your comment. We have added the explanation about the “VNRS” score in the legend of Figure 2.
The results of the study concluded that the combination of continuous lidocaine infusion and nefopam injection reduced acute postoperative pain compared with fentanyl-based PCA, while significantly reducing the incidence of postoperative nausea and vomiting (PONV) in women undergoing robot-assisted BABA thyroidectomy.
The limitations are presented fairly. The study used two prevention methods instead of a minimum of three as usual guides for postoperative nausea and vomiting (PONV) prevention, and no patient had PONV in the PO room. I would mention that the patients were females exclusively.
Response: Thank you. We agree with your opinion about a shortage of treatment compared to the guidelines of PONV prevention. In fact, it is not possible to prescribe the NK receptor antagonist for PONV in South Korea, so there is a practical difficulty in treating at least three medications prophylactically according to the guideline.
References are relevant, but it could be useful to discuss potential allergic reactions to anesthetic drugs and the evaluation of this risk. The authors can use as a reference:
Dumitru M, Berghi ON, Taciuc IA, Vrinceanu D, Manole F, Costache A. Could Artificial Intelligence Prevent Intraoperative Anaphylaxis? Reference Review and Proof of Concept. Medicina (Kaunas). 2022 Oct 26;58(11):1530. doi: 10.3390/medicina58111530. PMID: 36363487; PMCID: PMC9694532.
Response: Thank you for your comment. We have added that reference.
The article is a good work with clinical applicability. It needs more clinical trials to confirm the results (on male and female patients) and, of course, other variants of MMA, also.
Response: Thank you for your valid comment. We have revised the limitation section according to what you pointed out.
